# Short-Term Physiological Effects of a Very Low-Calorie Ketogenic Diet: Effects on Adiponectin Levels and Inflammatory States

**DOI:** 10.3390/ijms21093228

**Published:** 2020-05-02

**Authors:** Vincenzo Monda, Rita Polito, Annarita Lovino, Antonio Finaldi, Anna Valenzano, Ersilia Nigro, Gaetano Corso, Francesco Sessa, Alessio Asmundo, Nunzio Di Nunno, Giuseppe Cibelli, Giovanni Messina

**Affiliations:** 1Dipartimento di Medicina Sperimentale, Sezione di Fisiologia Umana e Unità di Dietetica e Medicina dello Sport, Università degli Studi della Campania “Luigi Vanvitelli”, 80138 Naples, Italy; vincenzo.monda@unicampania.it; 2Dipartimento di Scienze e Tecnologie Ambientali Biologiche Farmaceutiche, Università della Campania (Luigi Vanvitelli), 81100 Caserta, Italy; rita.polito@unicampania.it (R.P.); nigro@ceinge.unina.it (E.N.); 3Dipartimento di Medicina Clinica e Sperimentale, Università di Foggia, 71100 Foggia, Italy; annaritalovino@libero.it (A.L.); a.finaldi@gmail.com (A.F.); anna.valenzano@unifg.it (A.V.); gaetano.corso@unifg.it (G.C.); francesco.sessa@unifg.it (F.S.); giuseppe.cibelli@unifg.it (G.C.); 4Dipartimento di Scienze biomediche, odontoiatriche e delle immagini morfologiche e funzionali, sezione di Medicina Legale, Università di Messina, 98122 Messina, Italy; alessio.asmoundo@unime.it; 5Univesità del Salento, 73100 Lecce, Italy; nunzio.dinunno@unisalento.it

**Keywords:** very low-calorie ketogenic diet (VLCKD), adipose tissue (AT), adiponectin, cytokines, inflammatory diseases, visceral adipose tissue (VAT), C-reactive protein (CRP), lipid profile

## Abstract

Adipose tissue is a multifunctional organ involved in many physiological and metabolic processes through the production of adipokines and, in particular, adiponectin. Caloric restriction is one of the most important strategies against obesity today. The very low-calorie ketogenic diet (VLCKD) represents a type of caloric restriction with very or extremely low daily food energy consumption. This study aimed to investigate the physiological effects of a VLCKD on anthropometric and biochemical parameters such as adiponectin levels, as well as analyzing oligomeric profiles and cytokine serum levels in obese subjects before and after a VLCKD. Twenty obese subjects were enrolled. At baseline and after eight weeks of intervention, anthropometric and biochemical parameters, such as adiponectin levels, were recorded. Our findings showed a significant change in the anthropometric and biochemical parameters of these obese subjects before and after a VLCKD. We found a negative correlation between adiponectin and lipid profile, visceral adipose tissue (VAT), C-reactive protein (CRP), and pro-inflammatory cytokines such as tumor necrosis factor-α (TNF-α), which confirmed the important involvement of adiponectin in metabolic and inflammatory diseases. We demonstrated the beneficial short-term effects of a VLCKD not only in the treatment of obesity but also in the establishment of obesity-correlated diseases.

## 1. Introduction

Adipose tissue (AT) is a multifunctional organ involved in many physiological and metabolic processes. It is not only a site for energy storage but also an endocrine organ, composed of adipocytes, and it is also populated by several immune cells such as T lymphocytes and macrophages. As a result of excessive expansion of AT mass, a high-fat diet, lipolysis activation, and non-shivering thermogenesis recruit and activate numerous immune cells.

Through the production of adipokines and, in particular, adiponectin, AT is involved in many metabolic and inflammatory functions, as well as thermoregulation. Moreover, literature data demonstrate that obese people have a higher incidence of immune and autoimmune diseases [1,2]. In the obese condition, there is an accumulation of visceral adipose tissue (VAT) in the abdominal area of the body, which is extremely dangerous for health. Furthermore, white visceral fat adipocytes are particularly active in the release of adipokines, hormones involved in several metabolic and inflammatory processes, as well as in the normal homeostasis of many organs and tissues. Among these, adiponectin is the most abundant product from AT. It constitutes about 0.1% of total serum proteins. Adiponectin circulates as oligomers of different molecular weight, low molecular weight (LMW), medium molecular weight (MMW), and high molecular weight (HMW), which are the most biologically active. Through the direct or indirect release of this adipocytokine, visceral fat controls appetite and energy balance, immunity, angiogenesis, insulin sensitivity, and lipid metabolism [3].

Adiponectin has pleiotropic functions on different target tissues through the presence of its receptors, AdipoR1, AdipoR2, and T-cadherin in liver, muscle, and adipose tissue, where it positively affects homeostasis and metabolism of glucose and fatty acids [4,5]. As reported by Yamauchi et al., the expression of AdipoR1/R2 mediates increased AMP kinase and PPARα ligand activities, as well as fatty-acid oxidation and glucose uptake by adiponectin [5]. Adiponectin increases insulin sensitivity and reduces hepatic neo-glucogenesis. Furthermore, many studies reported that adiponectin could be an early marker assisting in the evaluation of the initial stages of a worsening glucose metabolism. The measurement of adiponectin concentration would aid in the identification of high-risk individuals with glycated hemoglobin (HbA1c) concentrations above a particular threshold, independently of serum glucose concentration. It is well known that glycated hemoglobin is a long-term marker of glucose metabolism; for these reasons, these two factors may be useful in the prevention of the initial stage of diabetes [6]. Among the inflammation markers, C-reactive protein (CRP) is an acute phase reactant marker of inflammation correlated with cardiac injury; in addition, it is strongly related to adiposity and insulin sensitivity. Literature data report that adiponectin and CRP serum levels are negatively correlated in type 2 diabetes and obesity [6]. Moreover, numerous studies, both in vitro and in vivo, also characterized the anti-inflammatory, anti-atherogenic, and anti-angiogenic effects of adiponectin [7,8,9].

The anti-inflammatory effects of adiponectin include both the suppression of pro-inflammatory cytokine production, such as TNF-α and IL-6, C protein, and growth factors, and modulation of the expression of anti-inflammatory cytokines such as IL-10 in monocytes and macrophages. On the other hand, TNF-α and other inflammatory markers (IL-6, C-reactive protein, SAA, tPA, MCP-1) and glucocorticoids suppress adiponectin production and regulate its levels. The anti-inflammatory cytokines, such as IL-10, positively correlate to adiponectin levels, and the production of this cytokine is stimulated by adiponectin [3,7]. Furthermore, this adipokine is involved in obesity outcome; in particular, it is strongly reduced in the serum of obese subjects. Moreover, it improves the oxidation of free fatty acids in muscle, preventing the growth of free fatty acids and triglycerides as a result of a high-fat diet. Numerous studies, both in vitro and in vivo, also characterized the anti-atherogenic and anti-angiogenic effects of this protein [7,8,9]. Furthermore, it is well known that adiponectin levels are strongly modified by diets such as the Mediterranean diet or the DASH diet [10].

In this scenario, caloric restriction represents one of the most important strategies against obesity. Caloric restriction reduces or slows the onset of diseases related to obesity, inducing a considerable weight loss and having beneficial and anti-inflammatory effects, reducing the production of free radicals, and favoring higher resistance to stress and prolonging lifespan. A very low-calorie ketogenic diet (VLCKD) represents a type of caloric restriction with very or extremely low daily food energy consumption (800 kilocalories per day or less). The VLCKD is a dietetic regimen that mimics fasting by restricting carbohydrates with a moderate increase in protein intake, which is proposed to achieve rapid weight loss. This diet is a particular ketogenic diet that was shown to be effective, at least in the short to medium term, as a tool to fight obesity [11]. This diet has various beneficial effects on numerous organs and tissues, inducing weight loss, reducing blood insulin levels, and cardiovascular risk factors, increasing mitochondriogenesis, and modulating neurotransmitter activity, as well as improving vascular density in the brain. The ketogenic diet also has an important role as a signaling mediator, driver of protein post-translational modification, and modulator of inflammation and oxidative stress. Several studies reported that VLCKD in the short term is able to reduce visceral adipose tissue and ameliorate lipid profile, in addition to reducing cardiovascular risk factors. [12,13]. The VLCKD is more effective in inducing weight loss compared with a standard low-calorie diet, presenting higher patient compliance; in fact, the ketone bodies increase satiety. The mechanism on which this satiety effect is based is complex and depends on the relationship that is established with several hormones and metabolites, mainly on a peripheral level [14]. The production of ketone bodies activates the ventromedial nucleus of the hypothalamus, which is directly related to satiety, and which varies throughout the day according to the intake of fats. As a consequence of this satiating effect, changes in body composition characterized by weight loss are produced, related to lower resistance to insulin and a low atherogenic lipid panel. Meanwhile, an increase in lean mass is shown; therefore, weight loss would be mainly based on a lower amount of body fat [14].

In light of these literature data, this study aimed to investigate the physiological effects of a VLCKD on anthropometric and biochemical parameters such as pro-inflammatory and anti-inflammatory cytokines, as well as on total adiponectin levels and its oligomeric profile. Monitoring was performed before and after eight weeks of VLCKD.

## 2. Results

### 2.1. Anthropometric and Biochemical Futures of VLCKD Obese Patients

The anthropometric and biochemical parameters of the VLCKD obese participants before and after eight weeks of nutritional intervention are reported in Table 1 weight, body mass index (BMI), fat mass (FM), and visceral adipose tissue (VAT) were statistically reduced in obese participants after the VLCKD. Moreover, biochemical parameters such as glycemic and lipid profiles were strongly ameliorated in these participants after the diet. Interestingly, the adiponectin levels in participants statistically increased after the diet, while CRP levels strongly decreased. As widely demonstrated, we found a sexual dimorphism in our study population. In particular, in VLCKD female obese participants, adiponectin levels were higher compared to male VLCKD obese participants both before and after the diet intervention (Table 2). For both males and females, we registered a strong increase in adiponectin levels after eight weeks of the VLCKD. Furthermore, adiponectin negatively correlated with VAT, with CRP (Figure 1), and with glycated hemoglobin and lipidic profile, but positively correlated with HDL-cholesterol (Figure 2). In addition, we found a strong reduction of TNF-α serum levels in participants after the VLCKD; on the contrary, IL-10 serum levels were statistically increased after the diet intervention. With regard to IL-6 serum levels, we found no strong differences between the two groups of participants (Table 1). TNF-α and IL-10 strongly correlated to adiponectin serum levels in participants before and after VLCKD (Figure 3). All these results showed that there is an improvement in anthropometric and biochemical parameters in a short period of diet intervention. 

### 2.2. Adiponectin Oligomerization State Analysis in Serum by Western Blotting Ting

The distribution of serum adiponectin in our VLCKD participants was analyzed by western blotting ting analysis (Figure 4). Three bands corresponding to HMW (≥250 kDa), MMW (180 kDa), and LMW (70 kDa) oligomers were evident in the serum from our participants; Figure 2A shows a representative blot of the different oligomers (HMW, MMW, and LMW) present in the serum of two VLCKD participants at baseline (T0) and after eight weeks of VLCKD (T1). The densitometric evaluation of the adiponectin oligomers in serum from all participants compared before and after eight weeks of nutritional intervention confirmed that HMW, MMW, and LMW oligomers increased after the dietary intervention (Figure 2B) (*p* < 0.05). In particular, HMW adiponectin oligomers, which are the most biologically active, were statistically increased in VLCKD participants after the period of diet intervention.

## 3. Discussion

The VLCKD is increasingly promoted as a strategy to fight obesity. Although the VLCKD is effective for weight loss and weight control, a comprehensive determination of its relationship with biochemical and physiological changes, in particular with the adipokines produced by AT, is still largely unexplored. This is the first study that evaluated total adiponectin serum levels using an ELISA test and its oligomeric profile using Western blotting analysis. Furthermore, the Western blotting analysis is a semi-quantitative method to support the previous results obtained by the ELISA test.

The main differences between low-calorie and ketogenic diets regard macronutrient composition. A low-calorie diet is a balanced diet characterized by 45–55% carbohydrates, 15–25% proteins, and 25–35% fat, with the additional dose of 30 g of fiber in the form of fruit and vegetables [15]. On the contrary, the ketogenic diet is low in carbohydrates (<50 g daily from vegetables) and lipids (only 10 g of olive oil per day) [16].

In this study, anthropometric and biochemical parameters in VLCKD participants before and after eight weeks of diet intervention were investigated, focusing on the effects of this diet on total adiponectin and its oligomeric profile. The results of the present study showed that the eight-week intervention with the VLCKD produced significant weight loss in our participants, decreasing pro-inflammatory cytokine production, increasing adiponectin serum levels, and improving metabolic profile. To support our results, several studies reported beneficial effects of a ketogenic diet [17,18,19,20]. In addition, VLCKD is advantageous in increasing satiety despite a negative energy balance, and sustaining basal energy expenditure despite body weight loss due to a sparing of fat mass [20,21,22]. Interestingly, as reported by Zhang et al., a ketogenic diet in combination with exercise reduced PPARγ and lipid synthetic genes, as well as enhancing the PPARα and lipid β-oxidation gene program in the liver compared to those in a ketogenic diet without exercise [23]. On the contrary, some studies reported that any beneficial effects are only transient [24]. In addition, Ellenbroek et al. supported that a long-term ketogenic diet (22 weeks) caused dyslipidemia, a pro-inflammatory state, signs of hepatic steatosis, glucose intolerance, and a reduction in beta and alpha cell mass, without weight loss in mice; however, the induction of ketosis and the response to ketosis in man and mouse are quite different, and 22 weeks is a very long period for a mouse that could be compared to several years in human beings [23,25]. The strength of this study is that, in a short period, the VLCKD changed the anthropometric and metabolic profile of our participants in a statistically significant manner. It is well known that VAT is dangerous to health, generating chronic low inflammation and leading to an imbalance in the function of adipose tissue; for these reasons, the inverse correlation between adiponectin and VAT and pro-inflammatory cytokines such as TNF-α is very important [26,27,28]. Gustafson and colleagues reported that, among the fat storage compartments in the body, VAT was found to be an important source of pro-inflammatory adipokines such as TNF-α and IL-6, and it was associated with an increased risk for atherosclerosis, more so than subcutaneous fat [27]. Moreover, significant changes in VAT, as well as in inflammatory and adipose tissue activity biomarkers, were reported, suggesting that the VLCKD, in the short term, can be considered very important in the deregulation of the balance between abdominal fat and the production of pro-inflammatory mediators. We also tested adiponectin, whose expression is strongly increased after eight weeks of VLCKD. Adiponectin is the most abundant circulating adipokine, and plasmatic levels, together with free fatty acids (FFA), are statistically inversely related to body fat, abdominal visceral fat, and glucose and lipid metabolism. This adipokine, through its HMW oligomers, which are the most biologically active, strongly increased in our participants after the diet intervention. Adiponectin exerts an anti-inflammatory effect and modulates insulin sensitivity by stimulating glucose utilization and fatty acid oxidation. On the contrary, elevated FFA was linked with the development of insulin resistance, defects in insulin secretion, nonalcoholic fatty liver disease, and metabolic syndrome [28]. As reported by Nigro et al., serum adiponectin levels are reduced in obese and diabetic subjects and are considered as a marker of various metabolic diseases, as well as of improvement of metabolic activity [29]. Furthermore, the results of the present study show a negative correlation between adiponectin and glycated hemoglobin; these findings, in agreement with Okoro et al., confirm the strong involvement of adiponectin in metabolic syndrome and in the establishment of type 2 diabetes [29,30,31]. However, the molecular mechanism underlying the cause-and-effect relationship between hypoadiponectinemia and insulin resistance is not yet fully clear [32]. Indeed, in vivo and in vitro studies suggested that adiponectin has an antidiabetic and hypoglycemic effect [33], activating hepatic insulin receptor and promoting pancreatic beta-cell function [33,34,35].

Data in the literature confirmed that the ketogenic diet is associated with increases in adiponectin in obese subjects [31,35]. Sherrier et al. hypothesized that the level of nutritional ketosis may be an important factor in the regulation of adiponectin expression, since ketones influence AMPK activity through adiponectin [36]. Furthermore, it is worth mentioning that adipose tissue is the target of many metabolically active factors, many of which induce the secretion of adiponectin. [36]. On the contrary, Garaulet et al. reported that adiponectin is related to protection against the metabolic syndrome but is not involved in the regulation of VLCD-induced improvement of insulin sensitivity [37]. In a chronic inflammatory state, such as obesity, the physiologic status of the cells presents changes in adipose tissue altering the production of adipokines [35]. In this scenario, another essential function of adiponectin and its HMW oligomers is their role in inflammatory and immune responses. We found a negative correlation between adiponectin and pro-inflammatory cytokines such as TNF-α; in fact, adiponectin is able to suppress nucleus NF-κB translocation and pro-inflammatory cytokine expression, including TNF-α, IL-1b, and IL-6 [38,39,40]. In addition, as we demonstrated, adiponectin positively correlates with IL-10 serum levels; in fact, it increases the expression of anti-inflammatory mediators, such as IL-10, and induces the polarization of anti-inflammatory M2 macrophages [41,42]. Furthermore, adiponectin enhances cold-induced browning of subcutaneous adipose tissue through M2 macrophage proliferation and promotes cell proliferation via the activation of serine/threonine-specific protein kinase Akt, consequently leading to beige cell activation [43,44,45]. Moreover, Tsuchida et al. suggested that adiponectin, through innate immune response-dependent mechanisms, can regulate insulin sensitivity and energy expenditure. On the other hand, adiponectin and its HMW oligomers are strongly involved not only in metabolic processes but also in inflammatory and immune responses [45]. In light of this evidence, the negative correlations that we found between adiponectin and lipid profile, VAT, CPR, and TNF-α confirmed the profound involvement of adiponectin in many metabolic and inflammatory diseases and, in parallel, also confirmed the beneficial short-term effects of VLCKD intervention not only in the treatment of obesity but also in the establishment of obesity-correlated diseases. Indeed, for its anti-inflammatory properties, the ketogenic diet is used as an adjuvant treatment in the cancer therapy. As suggested by Weber et al., a ketogenic diet creates an unfavorable metabolic environment for cancer cells and, thus, can be regarded as a promising adjuvant as a patient-specific multifactorial therapy [15]. The main limitation of this study is related to the small number of participants; contrariwise, the short time of the observation period can represent a strong point of the study. Indeed, the short period of VLCKD intervention has beneficial effects not only on metabolic rate but also on the inflammatory state, improving adiponectin and IL-10 levels, as well as reducing both TNF-α and IL-6 levels.

## 4. Materials and Methods

### 4.1. Participants

Twenty obese subjects (10 females, 10 males), aged between 20 and 60 years (mean 48 ± 10 years), were enrolled. The study took place at the Laboratory of Physiology, Department of Clinical and Experimental Medicine, University of Foggia. This study was performed in accordance with the Declaration of Helsinki and approved by the local ethics committee on 22 May 2018, n°440/DS. According to the current legislation in Italy, informed consent was signed by all participants, who were free to leave the study at any moment. Exclusion criteria were as follows: medical history positive for renal insufficiency, hyperuricemia, severe hepatic insufficiency, type 1–2 diabetes mellitus treated with insulin, atrioventricular block, heart failure, cardiovascular and cerebrovascular diseases, unbalanced hypokalemia, hypo-hyperthyroidism, chronic treatment with corticosteroid drugs, severe mental disorders, neoplasms, pregnancy, and lactation. All participants were highly motivated and none of them had any previous experience with low-carbohydrate or ketogenic diets.

### 4.2. Study Protocol

As previously reported, all participants underwent a general medical examination. We recorded age, height, weight, blood pressure, and laboratory tests, at baseline and after eight weeks [46,47].

During the study, dietary adherence was measured daily, between 2:00 and 4:00 p.m., by capillary blood ketone assessment, (GD40 Delta test strips, TaiDoc Technology Co., Taiwan). Nutritional ketosis was defined as a blood ketone (b-hydroxybutyrate) level >0.5 mmol/L. Body weight was recorded at the same time of the day, to the nearest 0.1 kg (SECA 711, Hamburg, Germany) with the participants wearing light clothing, and height was recorded to the nearest 0.1 cm (SECA 213, Hamburg, Germany). Fasting (12 h) blood samples were collected at 8:00 a.m., and blood samples were centrifuged with the resultant serum stored at −80 °C until use [47].

### 4.3. Anthropometric Measurements

Height, weight, BMI, and waist circumference of the 20 obese participants were recorded. Body weight was measured in a fasting state in the morning with a mechanical balance (±0.1 kg, SECA 700, Hamburg, Germany). BMI was calculated as body weight divided by height squared (kg/m^2^) with categories in accordance with the World Health Organization guidelines. To define participants with alterations in BMI and circumferences, the reference intervals normalized to the age of each participant were applied [47]. Body composition was estimated by DXA (Lunar Prodigy DXA, GE Healthcare, USA), via whole-body scan. VAT was quantified by subtracting subcutaneous fat from total abdominal fat, reported in grams, using the CoreScan software (GE Healthcare, Madison, WI, USA) [48].

### 4.4. Biochemical Parameters

Blood human samples were taken after obtaining the informed consent from the patients or control subjects in accordance with the tenets of the Declaration of Helsinki. The samples were collected after an overnight fast (12 h), and serum samples were collected. Serum albumin, insulin, C-reactive protein, glucose, total cholesterol, HDL and LDL cholesterol, and triglycerides were measured. Concentrations of total adiponectin in serum were measured via an enzyme-linked immunosorbent assay (ELISA) using a commercial kit (Elabscience, Houston, Texas, USA). Furthermore, we analyzed TNF-α, IL-10, and IL-6 serum levels via enzyme-linked immunosorbent assay (ELISA) using a commercial kit (BD Opt EIA for human TNF-α, IL-10, IL-6). All ELISA tests were performed in triplicate, and the protocols followed were according to the manufacturers’ instructions.

### 4.5. Western Blotting Ting Analysis

Serum samples from all participants were quantified for total proteins by the Bradford assay (Bio-Rad, Hercules, CA, USA), and 10 μg of total protein was heated in 1× Laemmli buffer at 95 °C for 10 min and loaded onto 10% SDS-PAGE gels as previously described [49]. The immunoblots were developed by ECL (Amersham Biosciences, Piscataway, NJ, USA) with the use of Kodak BioMax Light film, digitalized with a scanner (1200 dpi), and analyzed by densitometry with ImageJ software (https://imagej.nih.gov/ij/). Each sample was tested three times in duplicate.

### 4.6. Diet Intervention

The participants followed the VLCKD according to a commercial weight loss program (Lignaform, Therascience, 3, rue de l’Industrie, 98000 Principato di Monaco), previously reported in our published paper [47]. The characteristics of this diet are reported in Table 3.

### 4.7. Statistical Analysis

Statistical analyses were performed using the StatView software 5.0.1.0 (SAS Institute, Cary, NC, USA). All data are presented as means ± SD. A *p*-value ≤0.05 was used for statistical significance. Adiponectin, VAT, and CRP serum concentrations were correlated by Pearson’s or Spearman’s rho tests, according to data distribution. A *p*-value < 0.05 was considered statistically significant. Multiple comparisons on Western blotting ting experiments were performed.

## Figures and Tables

**Figure 1 ijms-21-03228-f001:**
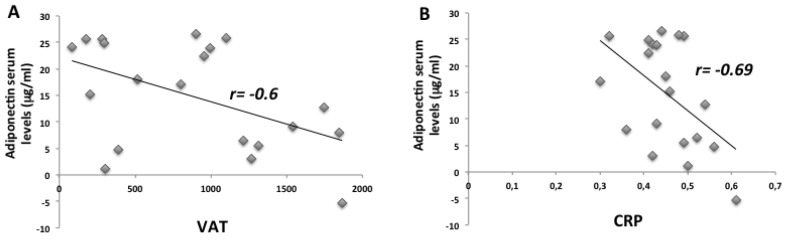
Adiponectin negatively correlated to VAT and CRP in VLCKD participants before and after diet. There is a negative correlation between VAT, adiponectin serum levels, adiponectin, and CRP serum levels in VLCKD participants (**A**,**B**).

**Figure 2 ijms-21-03228-f002:**
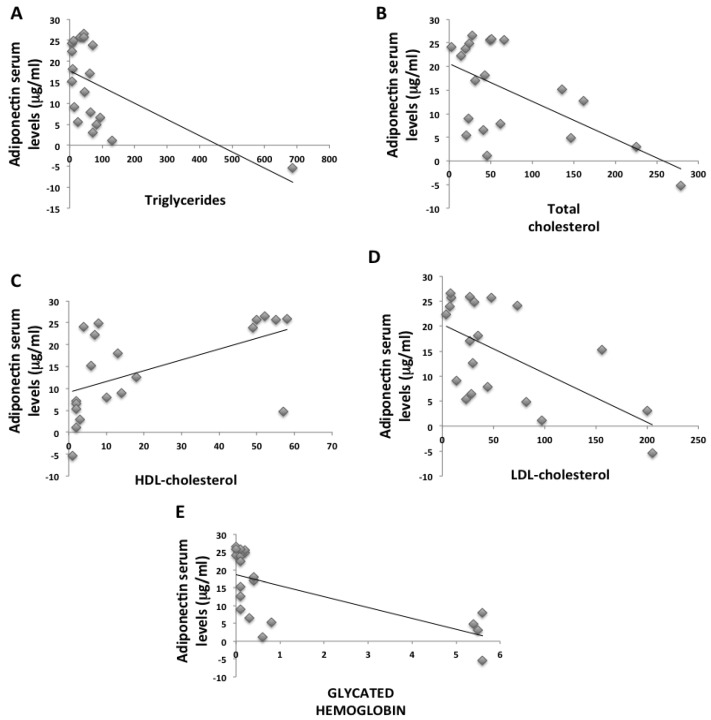
Adiponectin negatively correlated to glycated hemoglobin and lipidic profile but positively correlated to HDL-cholesterol in VLCKD participants before and after diet. There was a negative correlation between glycemic and lipid profile and adiponectin serum levels (**A**,**B**,**D**,**E**). On the contrary, there was a positive correlation between adiponectin and HDL-cholesterol (**C**).

**Figure 3 ijms-21-03228-f003:**
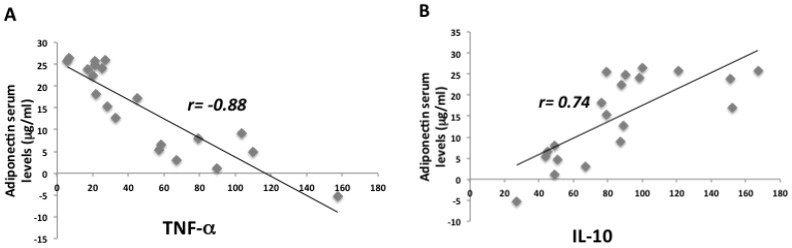
Adiponectin correlated to TNF-α and IL-10 serum levels in our participants before and after diet. There was a negative correlation between TNF-α and adiponectin serum levels (**A**). On the contrary, there was a positive correlation between adiponectin and IL-10 (**B**).

**Figure 4 ijms-21-03228-f004:**
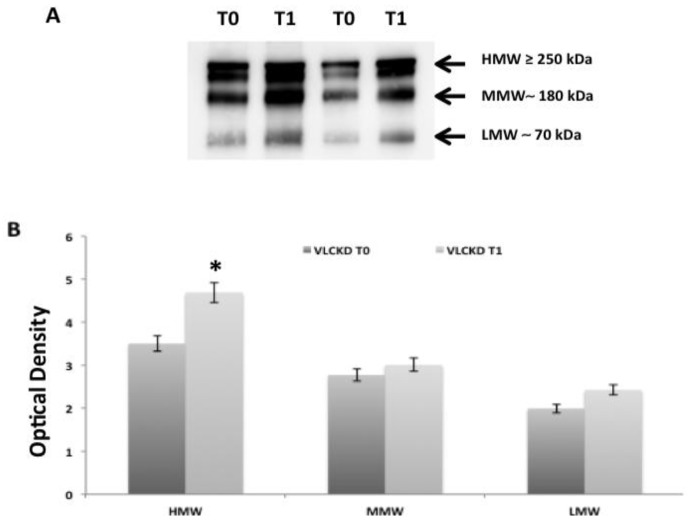
Improvement of adiponectin oligomeric profile after eight weeks of VLCKD. Adiponectin oligomers, analyzed by Western blotting ting, increased in the adiponectin from VLCKD participants after eight weeks of VLCKD (T1). (**A**) A representative blot of different oligomers of adiponectin (high, medium, and low molecular weight (HMW, MMW, and LMW)) in the serum of two VLCKD participants at baseline (T0) and after eight weeks of VLCKD intervention (T1). (**B**) Graphical representation of pixel quantization of all VLCKD participants at baseline (T0) and eight weeks after VLCKD intervention (T1).

**Table 1 ijms-21-03228-t001:** Body composition and biochemical features of very low-calorie ketogenic diet (VLCKD) of participants before and after weight-loss.

	VLCKD Obese Participants		*p*-Value
	T0	T1	
Sex male/female	10/10	10/10	
Age	48 ± 8.2	-	ns
Height (m)	1.67 ± 0.1		ns
Weight (kg)	91.33 ± 17.1	78.73 ± 13.3	<0.001
BMI (kg/m²)	32.19 ± 4.78	27.76 ± 3.6	<0.001
VAT (g)	1541.55 ± 141.6	927.79 ± 104.9	<0.001
FM (g)	39,208.77 ± 1432.5	27,377.03 ± 1217.4	<0.001
FFM (g)	48,789.57 ± 1712.3	48,093.68 ± 1670.6	ns
BMD	1225.57 ± 21.2	1229.31 ± 21.4	ns
Total cholesterol (mg/dL)	220.13 ± 50.7	173.91 ± 32.9	<0.05
HDL (mg/dL)	55.13 ± 11.1	47.76 ± 9.1	ns
LDL (mg/dL)	141.83 ± 36.4	107.57 ± 27.7	<0.05
Triglycerides (mg/dL)	135.54 ± 125.2	83.25 ± 26.1	<0.05
Glycemia (mg/dL)	96.68 ± 4.6	93.09 ± 3.3	<0.05
HGB (g/dL)	14.13 ± 1,3	13.83 ± 0.9	ns
Hba1c (%)	5.65 ± 0.3	5.38 ± 0.3	ns
Insulinemia (μU/mL)	10.53 ± 7.1	5.37 ± 3.7	<0.05
Uric acid (mg/dL)	4.86 ± 1.0	5.27 ± 1.1	ns
Total protein (g/dL)	7.30 ± 0.4	7.13 ± 0.4	ns
AST-GOT (U/L)	21.27 ± 5.9	23.31 ± 11.4	<0.05
ALT-GPT (U/L)	26.51 ± 14.8	26.06 ± 16.2	<0.05
Gamma GT (U/L)	31.19 ± 19.8	15.31 ± 5.4	<0.05
CRP (mg/mL)	0.89 ± 0.1	0.48 ± 0.1	<0.05
Adiponectin (μg/mL)	10.8 ± 1.2	25.55 ± 1.3	<0.001
TNF-α (pg/mL)	345 ± 6.5	278 ± 9.2	<0.05
IL-10 (pg/mL)	117 ± 7	168 ± 8.8	<0.001
IL-6 (pg/mL)	236 ± 4.4	232 ± 5	ns

Body mass index (BMI); visceral adipose tissue (VAT); fat mass (FM); fat-free mass (FFM); bone mineral density (BMD); hemoglobin (HGB); glycated hemoglobin (Hba1c); aspartate aminotransferase (AST); alanine aminotransferase (ALT); C-reactive protein (CRP); not significant (ns). Some of the data are the same as in Table 1 of our previously published paper [15].

**Table 2 ijms-21-03228-t002:** Adiponectin serum levels in male and female VLCKD obese participants.

	Adiponectin Levels (μg/mL)		*p*-Value
	Male	Female	
VLCKD obese participants T0	9.23 ± 0.7	12.44 ± 1.07	<0.05
VLCKD obese participants T1	23.67 ± 1.6	27.3 ± 1.33	<0.05

**Table 3 ijms-21-03228-t003:** VLCKD characteristics.

Fats (%)	43
Proteins (%)	43
Carbohydrates	14
Carbohydrates from vegetables (g/day)	<50
Total Kcal/day	700–900

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
