# Peer review of "Short-Term Physiological Effects of a Very Low-Calorie Ketogenic Diet: Effects on Adiponectin Levels and Inflammatory States"

_ijms, 2020, doi:10.3390/ijms21093228_

Round 1

Reviewer 1 Report

Title:

Short-term physiological effects of a very-low-calorie ketogenic diet: effects on metabolic and inflammatory states.

 Authors:

Vincenzo Monda, Rita Polito, Annarita Lovino, Antonio Finaldi, Anna Valenzano, Ersilia Nigro, Gaetano Corso, Francesco Sessa, Alessio Asmundo, Nunzio Di Nunno,Giuseppe Cibelli and Giovanni Messina.

Finding the molecular mechanisms underlying the beneficial whole-body changes that occur during weight-loss represents an important trend in research on obesity. From this point of view, I find the present study interesting; however, before publication, some significant issues should be clarified.

Major remarks:

Title

  • The title is slightly inaccurate since the Authors focus mainly on the VLCD-induced changes in adiponectin levels and present changes in other metabolic and inflammatory parameters as secondary to the variations in the adiponectin status. Shouldn’t the title be rephrased?

Introduction:

  • The inverse correlation between the weight loss and adiponectin levels (as well as between the increase in adiponectin levels and inflammatory parameters) is widely described. Therefore, the Authors should explain what the results of the present study add to the state of knowledge in this filed.
  • Please explain the selection criteria for the cytokines measured in the study protocol. For instance, what was the rationale to measure Il-6, not Il-1β levels?

Results

  • Table 1 –If the distribution of the particular variable was not normal, please provide medians and quartiles instead of means and SD.
  • Figures 1-3 Since there are only 20 values on each plot, do they refer to the moment before or after intervention?
  • Please improve the resolution of Figure 2.

Discussion:

  • Please, consider any potential limitations of the study: relatively small number of participants? A relatively short time of the observation period?
  • Please also discuss the studies which do not support the obtained results, e.g., Garaulet M, Viguerie N, Porubsky S, et al. Adiponectin gene expression and plasma values in obese women during very-low-calorie diet. Relationship with cardiovascular risk factors and insulin resistance. J Clin Endocrinol Metab. 2004;89(2):756–760. doi:10.1210/jc.2003-031495

Minor revisions:

 Whole text:

  • Please define the abbreviations as they occur for the first time in the text:

Lines 66-67 “On the other hand, CRP is strongly related to adiposity and insulin sensitivity.”

Lines 97-09 “The anthropometric and biochemical parameters of the VLCKD obese participants before and after 8 weeks of nutritional intervention are reported in Table 1: weight, BMI…”

Line 164 “Adiponectin is the most abundant circulating adipokine, and plasmatic levels, together with FFA…”

Line 195 “… and promotes cell proliferation by the activation of Akt.”

Table 1: add the table legend, where the abbreviations can be defined.

  • Please use the abbreviations consequently e.g., choose TNF-α or TNF-a.

Introduction:

  • Lines 56-58 “Through the direct or indirect release of these adipocytokines, the visceral fat controls appetite and energy balance, immunity, angiogenesis, insulin sensitivity, and lipid metabolism [3].” Does the expression “these adipokines” refer to the different adiponectin oligomers or different substances secreted by adipose tissue?
  • Lines 67-68 “Literature data report that adiponectin and CRP are two proteins negatively correlated to Type 2 Diabetes and obesity” please clarify this sentence. I suppose that the Authors mean: Literature data report that adiponectin and CRP serum levels are negatively correlated in Type 2 Diabetes and obesity.
  • Lines 71-72 “The anti-inflammatory effects of this adipokine include both the suppression of pro-inflammatory cytokine production, such as TNF-α and IL-6, C protein….” Did the Authors mean C-reactive protein (CRP)?
  • Lines 77-79: “Numerous studies both in vitro and in vivo have also characterized the anti-inflammatory, anti-atherogenic and anti-angiogenic effects of this protein [6–8].” Since the anti-inflammatory effects of adiponectin are described above, it would be reasonable to rephrase this sentence, e.g., “Numerous studies both in vitro and in vivo have also characterized the anti-atherogenic and anti-angiogenic effects of this protein [6–8].”
  • Lines 87-88: “This diet has various beneficial effects on numerous organs and tissue" – Should not be changed into: "This diet has various beneficial effects on numerous organs and tissues”?

Discussion:

  • Lines 156-159: “Gustafson and colleagues reported that among the fat storage compartments in the body, VAT has been found to be an important source of pro-inflammatory adipokines such as TNF, VAT has been found to be an important source of pro-inflammatory adipokines such as TNF-a and IL-6, and it has been associated with an increased risk for atherosclerosis, more so than subcutaneous fat [15].” Please, remove the repetition.
  • Lines 177-179: “Indeed, adiponectin has an antidiabetic and hypoglycaemic effect [21], activating hepatic insulin receptor and promoting pancreatic beta-cell function [22–24]." – please specify it these findings refer to the in vitro or to in vivo studies.

Author Response

We thanks the reviewer and we attached the reply letter. 

Reviewer 2 Report

The authors demonstrated that a ketogenic diet had beneficial effects on inflammatory and metabolic states.

The ms provides insufficient explanations and data.

From the data provided, it could not be stated which of the ketogenic diet or of the very low calorie diet is responsible for the loss of weight?

Introduction section provides insufficient information. The ideas need to be ordered (lanes 59-81). It is important to precise waht is the phenotype of mice deficient for AdipoR1 and AdipoR2? HbA1C is not determined nor is CRP. Therefore, it is not possible to understand why these two molecules are important is the demonstration. The VLCK diet is absolutely not detailed. What is this diet favoring? When first introduced, IL10 shuld be presented as anti-inflammatory. So it makes sens to study both TNFa and IL10

Mat&Meth. A table listing what is typical of VLVK diet?

Results: Data should also be presented per sex. It is well known that based on BMI an obese woman may have more subcutaneous fat than visceral fat. This is generally not true for men. Obese men have a high amount of visceral fat which is highly detrimental for metabolic health. Western-blotting data are not normalized. T1 appears systematically stronger than T0 whatever the band considered. A 10% SDS PAGE is not the best choice for studying proteins >150 KDa.

It is always difficult to conclude on Adiponectin plasma levels. In addition, males and females differ in their plasma levels.This is the reason why leptin is assayed. It is produced by adipocytes as a fonction of fat and the ratio of leptin to adiponectin is a very good marker.

Author Response

(The authors gave the same response as above.)

Round 2

Reviewer 1 Report

The authors have significantly improved the manuscript and clarified my doubts. Therefore, I find the paper suitable for publication in a present form.

Author Response

Thank you for the positive comment.

We have extensively checked the English, reviewing the manuscript.

Reviewer 2 Report

The authors have improved the ms although extensive english editing is required

However, two main points were not considered

1- what is the respective contribution of a low diet versus a ketogenic diet as the diet used is both a low and ketogenic diet. I understand the authors did not individually test the 2 diets but there is certainly data in the literature and this points ought to be discussed

2- what is the new contribution of the authors as it is written line 318 that a ketogenic diet will result in enhancing adiponectin secretionand it is also known that loosing weight results in enhanced adiponectin secretion

other points: Western blotting data are not completely convincing. Leptin measurement will be a plus

Author Response

Following your suggestion, we have improved the manuscript.

  • The authors have improved the ms although extensive english editing is required

We have extensively checked the English.

  • However, two main points were not considered

1- what is the respective contribution of a low diet versus a ketogenic diet as the diet used is both a low and ketogenic diet. I understand the authors did not individually test the 2 diets but there is certainly data in the literature and this points ought to be discussed

We thank the reviewer for this observation.

The low-calorie diet is comparable to the ketogenic diet in terms of calories. Indeed, both diets consist in 700/800 kcal/day, but these diets differ in micronutrient composition.

A low-calorie diet is a balanced diet based on the subdivision of macronutrients: 45-55% carbohydrates, 15-25% proteins and 25-35% fat; with recommended dose of 20-30 g fiber in the form of fruit and vegetables.  It is composed of whole grains such as bread, pasta, and rice, olive oil, vegetables, tomatoes, fresh fruit, cereals and 1 glass of milk every day; legumes 2 days/week; white meat 2 days/week; fish and seafood 3 days/week; 2 eggs 1 day/weeks

The ketogenic is a diet low in carbohydrates (<50 g daily from vegetables) and lipids (only 10 g of olive oil per day). The amount of high-biological-value proteins ranged between 0.8 and 1.2 g per each kg of ideal bodyweight, to ensure minimal body requirements and to prevent the loss of lean mass.  We have summarized this concept in the Discussion section (Line 184-188).

2- what is the new contribution of the authors as it is written line 318 that a ketogenic diet will result in enhancing adiponectin secretionand it is also known that loosing weight results in enhanced adiponectin secretion

The ketogenic diet ameliorates not only the anthropometric profile but also the biochemical profile of our obese subjects. In particular, the increase in adiponectin serum levels depends on weight loss as described in the literature. The new contribution of our study is that adiponectin not only plays a metabolic role increasing glucose uptake and ameliorating lipid profiles, but also this adipokine has an important role in inflammatory and immune responses. As we report in the paper adiponectin negatively correlates to pro-inflammatory cytokines such as TNF-α and IL-6 but positively correlates to anti-inflammatory cytokines such as IL-10, as we previously reported in lines 233-238.

In the text, line 180-183 we added this sentence:

“This is the first study that evaluated total adiponectin serum levels by ELISA-test and its oligomer profile by western blotting analysis. Furthermore, the western blot analysis is a semi-quantitative method to support the previous results obtained by the ELISA-test that is more quantitative”.

other points: Western blotting data are not completely convincing. Leptin measurement will be a plus.

We thank the reviewer for this suggestion. We did not perform leptin evaluation because the relationship between leptin and ketogenic diet has been described (Long-Term Effects of a Classic Ketogenic Diet on Ghrelin and Leptin Concentration: A 12-Month Prospective Study in a Cohort of Italian Children and Adults with GLUT1-Deficiency Syndrome and Drug Resistant Epilepsy. De Amicis R et al. Nutrients. (2019) ; Obesity and tumor growth: inflammation, immunity, and the role of a ketogenic diet. Wright C et al. Curr Opin Clin Nutr Metab Care. (2016).

Round 3

Reviewer 2 Report

The ms has been improved . I understand the authors did not compare the impact of a very low calorie diet versus a ketogenic diet. However, based on the literature the authors have to indicate why a ketogenic diet would be better than a very low calorie diet. Indeed, loosing weight is known by itself to improve metabolic parameters. But may be there are reports showing more beneficial effects of a ketogenic diet over a low calorie diet in terms of metabolic health and inflammatory status. These reports should be commented. Besides, several negative consequences have been published on diets favoring proteins and this has not been discussed.

Author Response

Following your suggestion, we have improved the manuscript.

  • The ms has been improved . I understand the authors did not compare the impact of a very low calorie diet versus a ketogenic diet. However, based on the literature the authors have to indicate why a ketogenic diet would be better than a very low calorie diet. Indeed, loosing weight is known by itself to improve metabolic parameters. But may be there are reports showing more beneficial effects of a ketogenic diet over a low calorie diet in terms of metabolic health and inflammatory status. These reports should be commented. Besides, several negative consequences have been published on diets favoring proteins and this has not been discussed.

    We thank the reviewer for these observations. We improved and clarified in the introduction the differences between VLCKD and low- calorie diet. Line 101-114 we reported:

“The ketogenic diet, has also an important role as signaling mediator, driver of protein post-translational modification, and modulator of inflammation and oxidative stress. Several studies reported that VLCKD in short terms is able to reduce visceral adipose tissue, ameliorate lipid profile, in addition reducing cardiovascular risk factors. [12-13.]. The VLCKD is more effective in inducing weight loss compared with a standard low-calorie diet, presenting higher patient compliance, in fact, the ketone bodies increase satiety. The mechanism on which this satiety effect is based is complex and depends on the relation that is established with several hormones and metabolites, mainly on a peripheral level [14]. The production of ketone bodies activates the ventromedial nucleus of the hypothalamus, which is directly related to satiety, and which varies throughout the day according to the intake of fats. As a consequence of this satiating effect, changes in body composition characterized by weight loss are produced, related to lower resistance to insulin and a low atherogenic lipid panel. Meanwhile, an increase in lean mass is shown; therefore, weight loss would be mainly based on a lower amount of body fat [14].”

 Also we improved the discussion section as we reported in

  • Line 207-218: “To support our results, several studies reported beneficial effects of ketogenic diet [20-23]. In addition, VLCKD is advantageous increasing satiety despite negative energy balance, and sustaining basal energy expenditure despite body weight loss due to a sparing of fat-mass [23-25]. Interestingly as reported by Zhang et al, ketogenic diet in combination with exercise reduced PPARγ and lipid synthetic genes, as well as enhancing PPARα and lipid β-oxidation gene program in the liver compared to those in ketogenic diet without exercise [26]. On the contrary, some studies reported that any beneficial effects are only transient [27]. In addition, Ellenbroek et al, supported that a long-term ketogenic diet (22 weeks) caused dyslipidemia, a pro-inflammatory state, signs of hepatic steatosis, glucose intolerance and a reduction in beta and alpha cell mass, without weight loss in mice, however, the induction of ketosis and the response to ketosis in man and mouse are quite different and 22 weeks is a very long period for a mouse that could be compared to several years in human beings [27-28].”.
  • Line 273-276: “Indeed, for its anti-inflammatory properties, the ketogenic diet is s used as an adjuvant treatment in cancer therapy. As suggested by Weber et al, ketogenic diet creates an unfavorable metabolic environment for cancer cells and thus can be regarded as a promising adjuvant as a patient-specific multifactorial therapy [20]. “